# Effect of Nanoparticles on Rheological Properties of Water-Based Drilling Fluid

**DOI:** 10.3390/nano13142092

**Published:** 2023-07-18

**Authors:** Yuan Lin, Qizhong Tian, Peiwen Lin, Xinghui Tan, Huaitao Qin, Jiawang Chen

**Affiliations:** 1Institute of Ocean Engineering and Technology, Ocean College, Zhejiang University, Zhoushan 316021, China; samylin@zju.edu.cn (Y.L.); 12134111@zju.edu.cn (Q.T.); 22134070@zju.edu.cn (P.L.); 22234073@zju.edu.cn (X.T.); qinhuaitao@zju.edu.cn (H.Q.); 2Hainan Institute, Zhejiang University, Sanya 572025, China; 3Petroleum Engineering Technology Research Institute, Sinopec Shengli Oilfield Company, Dongying 257001, China

**Keywords:** water-based drilling fluid, nanoparticles, rheological properties, filtration property

## Abstract

Nano-water-based drilling fluids (NWBDFs) are prepared using nano-copper oxide (CuO) and multiwalled carbon nanotubes (MWCNTs) as modification materials. The effects of the temperature and concentration of the nanoparticles (NPs) on the rheological properties are studied using a rotational rheometer and viscometer. Also, the influence of two NPs on the filtration properties is studied using a low-pressure and low-temperature filtration apparatus, as well as a scanning electron microscope (SEM). It is found that MWCNTs with a concentration of 0.05 w/v% have the most obvious influence on the NWBDFs, which improve the stability of the gel structure against temperature and also decrease the filtration rate. Finally, a theoretical model predicating the yield point (YP) and the plastic viscosity (PV) as a function of the temperature considering the influence of the NPs is developed based on DLVO theory.

## 1. Introduction

Water-based drilling fluid (WBDF) has the advantages of low cost and ecofriendliness, which, therefore, results in its wide use in the oil and gas industry. The aqueous bentonite dispersion, as a clay dispersion with strong interparticle interactions, is the major component of WBDF. The rheological behavior of various clay dispersions has been well studied [1,2,3,4,5]. Because of the heterogeneous charging conditions at the edge and face, clay platelets in the dispersion have strong electrostatic interactions with each other, leading to a face-to-face or face-to-edge structure, based on which large clusters can form, giving rise to gelation of the system in the static state. Consequently, clay dispersion shows a significant yield behavior—below a critical stress defined as the yield stress, the system is solid-like. With an applied stress above the yield stress, the system becomes liquid-like and shows a significant shear-thinning behavior, i.e., the viscosity decreases with an increasing shear rate applied to the system, which is due to the disaggregation and orientation of the structure from the micro- to the nano-scale [6,7,8,9]. The rheological behavior of the clay dispersions is sensitive to the change in the clay’s concentration, ionic strength, and temperature, as well as the pH of the system [4,10,11,12].

Because of the thermal sensitivity of the bentonite dispersion, which is the base fluid of a WBDF, in deep-sea oil and gas exploration, the use of WBDF encounters the challenge of the significant alteration of the rheological behavior due to the change in the local temperature from the sea surface to the underground layer below the seabed, going through a process that first decreases then increases with increasing depth, starting at the sea’s surface. The variation of the overwhelming temperature leads to difficulty in controlling the flow property in the well [13], including both the yield behavior and the viscosity. Therefore, it is essential to improve the properties of the WBDF for a more efficient drilling process in the deep-sea oil and gas industry.

The rheological property of the WBDF can be improved using various kinds of additives, including nanocellulose [14], CuO/ZnO [15], MWCNTs [16], and SiO_2_ [17]. In recent years, a number of studies have been conducted to research the effect of NPs as treatment agents to improve the rheological properties of drilling fluids, as shown in Table 1. In general, it is agreed that NPs have the ability to improve the rheological performance, such as altering the shear thinning of the fluids and the yield stress, as well as increasing its rheological stability against the change in temperature and pressure. Accordingly, the filtration performance of drilling fluids can be improved. Nevertheless, there are contradictory conclusions concerning the effect of NPs on the viscosity of drilling fluids in previous studies, and the nanoscale mechanism altering the flow behavior using nanoparticles is hazy at the present stage.

In this paper, nano-water-based drilling fluids (NWBDFs) were prepared by adopting spherical copper oxide (CuO) nanosized particles and nonspherical multiwalled carbon nanotubes (MWCNTs) as modification additives. The influence of the NP concentration and temperature on the rheological property of the NWBDF was evaluated experimentally using an advanced rotational rheometer, as well as a traditional viscometer. Theoretical models are proposed based on the investigation of the mechanism.

## 2. Experiment

### 2.1. Preparation Method for NWBDF

The nano-water-based drilling fluids (NWBDFs) used in this study were prepared by adding nano-fluids into the water-based drilling fluids (WBDFs). The volume fraction of the nano-fluids was 10%. For the preparation of the WBDFs, we used a dispersion of OCMA-grade bentonite as the base fluids, xanthan gum (XG) as the viscosifier, and low-viscosity polyanionic cellulose (PAC-LV) as the filtrate reducer. Potassium chloride (KCl) was used as the shale inhibitor. Adding XG can considerably increase the viscosity of drilling fluids system [34].Here, the concentration of XG was unique in each sample in order to study purely the effect of the NPs. Table 2 shows the dosage of all of the components for the preparation of the WBDFs used in this study, according to the work of Novara et al. [31]. The bentonite powder was slowly poured into the deionized water, which at the same time was homogenized using a stirrer at 1000 rpm. The KCL, XG, and PAC-LV were also homogeneously mixed, after which they were added to the prepared bentonite dispersion. The mixture was stirred continuously for 2 h, leading to the WBDF sample used in our study.

For the nano-fluids, both nanosized copper oxide particles (CuO, average diameter: 25 nm, purity ≥ 99.99%) and multiwalled carbon nanotubes (MWCNTs, purity ≥ 95%, inner diameter: 3–5 nm, outer diameter: 8–15 nm, and length: 3–12 μm) were adopted as additives for the drilling fluids. Polyvinyl pyrrolidone (PVP) was used as a dispersant for a more homogeneous mixing of the NPs in the nano-fluids. SEM images of the two NPs are shown in Figure 1. It can be seen that the copper oxide nanoparticles were spherical particles, while the multiwalled carbon nanotubes were slender tubes. The detailed components of the nano-fluids are listed in Table 3. Ten nano-fluids with different concentrations of CuO and NWCNTs were adopted in this study in which the first group, as shown in Table 3, was the control group without the addition of NPs. In the preparation of the nano-fluids, we used an ultrasonic agitator to disperse the NPs in the mixture for 1 h. The prepared NWBDFs samples were rested for 20 h before the experiment was applied. The detailed mixing process is shown in Figure 2.

### 2.2. Experimental Method

In the rheological experiment, we adopted a DHR-1 stress-controlled rotational rheometer from TA Instruments. A DHR-1 rotational rheometer from TA Instruments with a cone-plate geometry was used in the rheological measurements. The diameter of the plate was 40 mm, with the cone angle of 1.983°. The cone was truncated at 48 μm from the vertex. The shear rate control mode was used to measure the rheological curves of the NWBDF at a temperature of 3 °C, 10 °C*,*
20 °C*,*
30 °C*,*
40 °C, and 60 °C, and the test was repeated five times for each temperature. The experimental procedure was as follows: a preshearing treatment at a shear rate of $1022\;s^{-1}$, and a duration of 10 s applied to each sample. The sample was then rested for 30 s, after which a ramp-up shear experiment with a stepwise increment of the shear rate from 0.1 to $1200\;s^{-1}$ was carried out. The shearing time in each shear step was 10 s in order for the system to achieve an equilibrium shearing state before the rheological data were recorded [7]. Also, the frequency sweep test at a small amplitude oscillatory shear (SAOS) was carried out for an investigation of the influence of the two NPs on the gel structure of the drilling fluids. In the experiment, a stress control mode was adopted with a stress amplitude of 0.1 Pa, which is well below the yield stress of the sample and, therefore, it was considered that the gel structure would not be altered by the SAOS test. An anti-volatilize solvent trap was adopted at a temperature above 30 °C so that the sample was tested under the saturated vapor pressure of the solvent. This allowed for the property of the aqueous sample to be stable over a long experimental period of time.

The relationships among the shear stress measured from the rotational rheometer and the apparent viscosity (AV), plastic viscosity (PV), and yield point (YP) are expressed as [35]:(1)$$\mathrm{AV}=\frac{\tau_{1022}}{1022}\times 1000\;(\mathrm{mPa}\cdot s)$$
(2)$$\mathrm{PV}=\frac{\tau_{1022}-\tau_{511}}{1022-511}\times 1000\;(\mathrm{mPa}\cdot s)$$
(3)$$\mathrm{YP}=\tau_{1022}-\frac{\mathrm{PV}}{1000}\times 1022\;(\mathrm{Pa})$$
where $\tau_{511}$ and $\tau_{1022}$ are the corresponding shear stress at a shear rate of $511\;s^{-1}$ and $1022\;s^{-1}$, respectively. As the rheological curves at each temperature and concentration of the NPs were repeated five times, the mean values of the AV, PV, and YP of the above rheological parameters were taken as the mean values of the five repeated experiments. The rheological properties of the NWBDFs were also tested using a traditional viscometer, which has been widely adopted for rheological analyses of drilling fluids, including $G_{10s}\;$(i.e., shear stress measured after resting for 10 s) and $G_{10\min}$ (i.e., shear stress measured after resting for 10 min) at room temperature (25 °C). The gel strength data were averaged over 10 repeated tests.

As for the filtration performance of the NWBDFs, an API standard filtration test of the NWBDFs was carried out using a medium pressure filtration apparatus.

## 3. Results and Discussion

### 3.1. Analysis of the Rheological Behavior under Steady Shear

With the adoption of the controlling shear rate mode, the change in the shear stress with the shear rate for the drilling fluids with and without NPs at different temperatures can be obtained, as shown in Figure 3. A similar trend can be observed for the three drilling fluids, showing a non-Newtonian shear thinning behavior at large shear rates. Furthermore, the NWBDF with 0.15 wt% MWCNTs was less sensitive to the temperature compared to the other two drilling fluids.

According to Equations (1)–(3), the apparent viscosity (AV), plastic viscosity (PV), and yield point (YP) of the NWBDFs with various concentrations of nanoparticles (NPs) are calculated as shown in Figure 4 and Figure 5. Obviously, the AV gradually decreased with an increasing temperature (Figure 4b and Figure 5b). Furthermore, when the concentration of the added NPs was less than 0.025 w/v%, there was basically no difference in the AV among the samples using nano-CuO and MWCNTs. When the NP concentration was greater than 0.025 w/v%, a difference was present. For the CuO-NWBDF, with an increase in the concentration of NPs, the AV first increased, reaching a peak value of 0.05 w/v%, and then decreased to a trough value of 0.10 w/v%, after which the AV further increased [20]. This phenomenon was observed at all temperatures investigated. On the other hand, for the MWCNT-NWBDF, with an increasing concentration of NPs, the AV increased, reaching the maximum value of 0.075–0.10 w/v%, and then it decreased. It can be noted that for samples using both NPs, the AV generally increased compared to the control group. Furthermore, the AVs investigated here were all sensitive to the temperature, similar to the sample without NPs [36].

Similar to the behavior of the AV, the PV decreased as the temperature increased. As shown in Figure 4b and Figure 5b, the PV was smaller than the AV and had a different trend with an increasing concentration of NPs. At a low temperature (3 °C, 10 °C, and 20 °C), the two NWBDFs showed almost opposite trends. With increasing NPs, for the NWBDF with CuO NPs, the PV first increased and then decreased, which further increased above 0.1 w/v%. While for the NWBDF with the MWCNT NPs, the PV first decreased and then increased, which further decreased above 0.1 w/v%. When the temperature increased to a high level (30 °C, 40 °C, and 60 °C), the behaviors of the PV with both NPs were similar to the AV, as shown in Figure 4a. 

As shown in Figure 4c and Figure 5c, the yield point (YP) decreased with the increase in temperature, similar to the behavior of the control group. With the increase in the concentration of the MWCNT NPs, a significant increase in the YP can be observed at the stage when the concentration of the NPs was lower than 0.075 *w*/*v*%, beyond which a decrease in the YP with an increasing concentration in present. The evolution of the YP become less sensitive to the addition of the MWCNT NPs with an increase in temperature. On the other hand, for the CuO NPs, a change in the YP with the NP concentration was less obvious. With an increasing concentration of NPs, the YP increased and peaked at approximately 0.05 *w*/*v*%, and then it decreased to a local minimum of 0.1 *w*/*v*%, after which the YP increased once more.

The shear thinning behavior of the drilling fluids was evaluated using the ratio of the YP to PV (YP/PV), with a larger YP/PV accounting for a more obvious shear thinning. Figure 4d and Figure 5d show the variation in YP/PV for the two NWBDFs at different temperatures. It can be seen that the YP/PV increased with an increasing temperature, indicating that with a higher temperature, a stronger non-Newtonian behavior of the NWBDFs was expected. It can be seen that the YP/PV was more sensitive to the addition of the MWCNTs at a NP concentration lower than 0.075 w/v% and at a lower temperature (3 °C, 10 °C, and 20 °C). On the other hand, the change in the YP/PV was not obvious with the addition of the CuO NPs at a low temperature range, the effect which, however, became more obvious when the temperature was above 30 °C. From the above investigation, we may deduce that the MWCNTs were more efficient at enhancing the shear thinning of the WBDF compared to the CuO NPs.

The gel strength of the drilling fluids measures the strength of the structure formed in the drilling fluids in the static state [37]. The gel strength was evaluated using two parameters, namely, $G_{10s}$ and $G_{10\min}$, which were used to measure the thixotropy of the drilling fluids [38], i.e., the rate of recovery of the gel structure [38]. Figure 6 shows the change in the gel strength of the two NWBDFs with the concentration of the NPs measured with the viscometer. It can be seen that change in the gel strength measured using the viscometer was similar to the variation in YP, as shown in Figure 4 and Figure 5. The change in the gel strength was more sensitive to the addition of the MWCNTs compared to the CuO. The inset shows the change in $G_{10\min\;}\mathrm{and}\;G_{10s}$ with the concentration of the NPs. It is observed that the thixotropy of the WBDF showed an increasing trend with the increasing concentration of NPs adopted in this study.

### 3.2. Analysis of the Rheological Behavior during a SAOS Experiment

A small amplitude oscillating shear (SAOS) experiment was used to investigate the effect of the NP concentration on the gel structure of the NWBDFs. The storage modulus, G′, loss modulus, G″, and phase angle, δ, of the two NWBDFs at different temperatures and NP concentrations were studied. Figure 7 indicates the evolution of the G′, G″, and δ at a frequency of 1 Hz. It can be found that the modulus of the WBDF was more sensitive to the addition of the MWCNTs compared to the CuO additive. The behavior of the dynamic modulus at the low frequency range, which characterizes the structure of the sample in the gel state, was similar to the evolution of the YP, as well as the gel strength, both of which evaluated the strength of the gelation in the sample. Furthermore, it can be observed that the samples with MWCNTs were more solid-like and, accordingly, had a larger G′ compared to the one with CuO. This was also confirmed with the YP and the gel strength of the system, which was obviously larger for the samples with MWCNTs, as shown in Figure 5 and Figure 6. 

Figure 8 shows the change in the modulus at 1 Hz with the temperature. The dynamic modulus for the samples with and without NPs both changed with the alteration of the temperature. For the samples without NPs and with CuO NPs, the behavior was similar. Both G′ and G″ decreased with an increasing temperature. The phase angle increased with the temperature, indicating that with an increasing temperature, the samples were more liquid-like. It is noted that the addition of CuO NPs suppressed the variation in the modulus with the temperature. On the other hand, for the samples with MWCNTs, it can be observed that G′ seemed not to be sensitive to the temperature, while G″ decreased with an increasing temperature, giving rise to a decrease in the δ with an increase in the temperature. This means that the system was more solid-like at a higher temperature, which cannot be reflected by a YP that decreases with temperature, as shown in Figure 5. This was probably due to the decreasing viscosity of the liquid phase with an increasing temperature. 

In summary, it was observed through the SAOS experiment that the thermal stability of the rheological properties of the WBDF in the static state can be improved through using both CuO and MWCNT additives in which the MWCNTs seem to be a much more efficient additive compared to the CuO NPs.

### 3.3. Analysis of the Microstructure

Figure 9 indicates the change in the API standard filtration of the WBDF with the concentration of NP additives, which show a similar manner. Both filtrations first decreased and then increased with an increasing concentration of NPs. For the CuO NPs, the minimum filtration was reached at 0.1 w/v%, while for the MWCNTs, the minimum filtration was achieved at a concentration of 0.05 w/v%. Obviously, it can be found that the MWCNTs were also a more efficient additive to reduce the filtration compared to the CuO NPs given the proper concentration. While with a further increase in the concentration of the MWCNTs, a more obvious negative effect on the filtration could be observed compared to the CuO NPs. This agrees with the finding by Ismail et al. [22]. It was noted that when the concentration of NPs was lower than 0.05 w/v%, the filtration decreased with an increase in the concentration and, accordingly, the YP increased. Meanwhile, that critical concentration of NPs accounting for the minimum filtration was also close to the critical concentration referring to the maximum PV, as shown in Figure 4b and Figure 5b. This indicates that a similar mechanism may be proposed accounting for both the enhancement of the gel structure and reduction of the filtration of the mud cake.

Figure 10 and Figure 11 show the filter cakes obtained from the filtration experiments. The structures of the mud cakes were investigated using a scanning electron microscope (SEM) after a drying process, as shown in Figure 12, Figure 13 and Figure 14. For the pure WBDF without NPs (Figure 12), it can be found that the filter cake was mainly composed of clusters formed by stacks of bentonite platelets. The addition of NPs significantly changed the interaction of the clay platelets. It can be observed that, in practice, the NPs were not homogeneously distributed in the sample if the local concentration was too high, as is evidenced in Figure 13 and Figure 14 (both 0.10 w/v% NPs were added), which provided us with a hint to propose a configuration of NPs at different concentration levels. For the spherical CuO NPs, as shown in Figure 13a,b, the well-dispersed nanosized particles may fit into the space between the clay platelets, as well as the space among clusters of clay platelets, as illustrated in Figure 15a. This enhances the gel structure of the WBDF and, consequently, reduces the rate of filtration. While with a further increase in the NP concentration, as shown in Figure 15b and evidenced in Figure 13c, the CuO particles themselves may form aggregations, which hinders the electrostatic interaction among bentonite platelets and, thus, weakens the gel structure. The formation of aggregates of CuO particles also leads to a looser structure in the mud cake, which increases the filtration rate. The phenomenon proposed is similar to the finding by Dejtaradon et al. and Saboori et al. [28,29]. For MWCNT additives, as shown in Figure 1b, because of the large length-to-diameter ratio of theses tubes, with a length scale that reaches 10 μm, it is considered that a low fraction of MWCNT particles could more effectively promote the connection among clusters of bentonite dispersions in the WBDF, as shown in Figure 14a and Figure 16a, leading to a significant enhancement of the gel structure, which also reduces the filtration of the consequent mud cake. While similar to the CuO NPs, a high concentration of MWCNTs may lead to loose packed aggregations of nanotubes that hinder the further enhancement of the gelation of the system, which, on the contrary, greatly loosen the structure (Figure 14b and Figure 16b), giving rise to an obvious increase in the filtration rate. 

### 3.4. Theoretical Model

In a previous study, the model predicting the yield stress that evaluates the gel strength of clay dispersions was developed from DLVO theory [11]. The yield stress, $\tau_{y}$, of the gel system can be estimated using the following equation [39]:(4)$$\tau_{y}\propto -\frac{\varphi^{2}}{a^{2}}W^{\prime }\left(D_{0}\right)$$
where a is the effective radius of the suspending particles, φ is the volume fraction of the clay particles in the system, D is the distance between clay particles, and $D_{0}$ is the distance between clay particles at the yield point of the system. W(D) is the interaction potential between clay particles, which can be expressed in the following equation considering that both the van der Waals interaction and the electrostatic interaction between particles are prominent [40]:(5)$$W\left(D\right)=W_{vdw}\left(D\right)+W_{e}\left(D\right)$$
where $W_{vdw}\left(D\right)$ is the van der Waals potential, and $W_{e}\left(D\right)$ is electrostatic potential. It is estimated that:(6)$$W_{vdw}=-\frac{aA_{H}}{12D}$$
in which $A_{H}$ is the Hamaker constant. Considering that the electrostatic force between clay particles is dominated by the repulsive interaction [41], the electrostatic potential is expressed as:(7)$$W_{e}=2\pi \varepsilon_{0}\varepsilon a\psi_{s}^{2}\ln\left[\frac{1}{1-\exp(-\kappa D)}\right]$$
where $\varepsilon_{0}$ is the vacuum dielectric constant; ε is the relative dielectric constant of the liquid phase and is a function of temperature, T. $\psi_{s}$ is the electrostatic potential of particle surface. κ is the reciprocal of the debye length, which is also a function of temperature, T. Combination Equation (4) with Equation (7), the yield stress of the clay gel system can be deduced as
(8)$$\tau_{y}\propto -\frac{\varphi^{2}}{a}\left[\frac{A_{H}}{12D_{0}^{2}}-\frac{2\pi \varepsilon_{0}\varepsilon \psi_{s}^{2}\kappa }{\exp(\kappa D_{0})-1}\right]$$

As the pH of the WBDF is approximately 9, the surface and edge of the clay particles will be negatively charged [42], and the model does not consider the face–edge electrostatic attraction of the clay particles. The relationship between clay volume fraction, φ, and the particle spacing, $D_{0}\;$, can simply be expressed as $\varphi =\delta_{p}/\left(\alpha D_{0}+\delta_{p}\right)$, where $\delta_{p}$ is the thickness of clay particles, and α is the correction factor [41]. In addition, we propose that the YP of the NWBDF is proportional to the yield stress of the gel system $\left(\mathrm{YP}\propto \tau_{y}\right)$, both of which reflect the gel strength of the system. Consequently, the theoretical model of the YP of the NWBDF on the temperature, T, can be written as:(9)$$\mathrm{YP}\left(T\right)=-\frac{\varphi^{2}}{{\left(\varphi^{-1}-1\right)}^{2}}\tau_{vdw}+K_{e}\frac{\varphi^{2}\kappa \left(T\right)\varepsilon \left(T\right)}{\exp\left[\delta \prime \left(\varphi^{-1}-1\right)\kappa \left(T\right)\right]-1}$$
where $\tau_{vdw}\propto A_{H}12a\delta \prime^{2}$ is related to the van der Waals force. $K_{e}\propto 2\pi \varepsilon_{0}\psi_{s}^{2}/a$ is the related to the electrostatic force; $\delta^{\prime }=\delta_{p}\;/\alpha$. The reciprocal of the Debye length, κ, can be estimated using the following equation [39]:(10)$$\kappa =\sqrt{\frac{2z^{2}e^{2}N_{A}c}{\varepsilon \varepsilon_{0}kT}}{\;[m}^{-1}]$$
where z is the valence state of the salt, for KCl, z=1; $e=1.6\times 10^{-19}\;C$ is the basic charge quantity; $N_{A}=6.02\times 10^{23}{\;\mathrm{mol}}^{-1}$ is Avogadro’s constant; c=0.5174 mol/L is the salt concentration in the system; $\varepsilon_{0}=8.85\times 10^{-12}{\;C}^{2}/\left(J\cdot m\right)$ is the vacuum dielectric constant. $k=1.38\times 10^{-23}\;J/K$ is the Boltzmann constant; T is the absolute temperature of the solution in units of K. Consequently, Equation (10) can be rewritten as:(11)$$\kappa \left(T\right)=\frac{11.427}{\sqrt{T\cdot \varepsilon \left(T\right)}}{\;[\mathrm{nm}}^{-1}]$$

Substituting Equation (11) into Equation (9), the models describing YP as a function of the particle concentration, as well as the temperature, can be obtained.
(12)$$\mathrm{YP}\left(T\right)=-\frac{\varphi^{2}}{{\left(\varphi^{-1}-1\right)}^{2}}\tau_{vdw}+\sqrt{\frac{\varepsilon \left(T\right)}{T}}\frac{11.427K_{e}\varphi^{2}}{\exp\left[11.427\delta^{\prime }\left(\varphi^{-1}-1\right)/\sqrt{T\cdot \varepsilon \left(T\right)}\right]-1}$$

In Equation (12), the clay volume fraction, *φ*, can be calculated according to $\varphi ={\left[\left(\rho_{c}/\rho_{w}\right)\left(c_{1}{}^{-1}-1\right)+1\right]}^{-1}$, where $\rho_{c}=2.6\;g/{\mathrm{cm}}^{3}$ is the density of the clay, $\rho_{w}=1\;g/{\mathrm{cm}}^{3}$ is the density of the water, and $c_{1}$ is the mass fraction of the clay, so φ=0.0194. Parameters $\tau_{vdw}$, ${K^{\prime }}_{e}$, and $\delta^{\prime }$ can be determined by fitting to the experimental data, and the results are shown in Table 4. The mathematical relationship of the relative dielectric constant of water with the absolute temperature, T, can be described as:(13)ε(T)=174.7218−0.32213T.

The above equation is an empirical model obtained by fitting the experimental data from Dass [43], as shown in Figure 17, which shows that ε is approximately linear with the temperature.

Using Equations (12) and (13) and noting that φ=0.0194, the change in YP with the temperature at different concentrations of NPs was fitted, as shown in Figure 18. The parameters $\tau_{vdw}$ and $K_{e}$ were adopted for fitting and are listed in Table 4, which varied with the concentration of NPs. The parameter $\delta^{\prime }$, which is related to the thickness of the clay particles and the correction factor α, was considered to be constant. The fitted correlation coefficients are also listed in Table 4. The correlation coefficients were basically around 0.9, indicating that the model proposed is reasonable. It can be observed that the model developed well and predicted the variation in the YP with change in the temperature considering the NPs.

The above developed model can also be used to predict the behavior of the PV. For drilling fluids, PV is used to measure the internal friction in a laminar flow, which, for clay dispersions, is considered to also be dominated by the interparticle interaction accounting for the gelation of the system [8]. Therefore, a change in PV should be similar to YP, which reflects the strength of the interaction force among clay particles. Here, we assume a power law relationship between PV and YP, expressed as:(14)$$\mathrm{PV}=P\cdot {\left(\mathrm{YP}\right)}^{m}$$

Equation (14) is used to fit the variation of PV with YP with changing concentrations of NPs, as shown in Figure 19, and the values of the parameters P and m obtained by fitting are shown in Table 5. The fitted correlation coefficients are also listed in Table 5. The correlation coefficients are basically 0.9, which also shows that the model proposed is reliable.

Using Equations (12)–(14), the variation in the PV with the temperature, T, and comparing the model with the experimental value, as shown in Figure 20, it can be found that the fitting was good, except at a temperature of 60 ℃, where deviation between the model and the experimental data can be observed. This reflects that at a high temperature, and in flow, the dominant interaction between particles may not be identical to that at a static state accounting for the YP.

## 4. Conclusions

The modification of the rheological properties and filtration performance of the WBDF through the addition of two NPs (nano-CuO and MWCNT) were studied. It was found that nonspherical MWCNT NPs could more efficiently change the property of WBDF at an optimized concentration of 0.05 w/v%, which improved the stability of the gel structure against the temperature and also decreased the filtration rate. The mechanism of the modification of the structure and the filtration was proposed. Furthermore, based on DLVO theory, a theoretical model predicting the YP and PV with temperature considering the influence of the NPs was developed. The model showed good agreement with the experimental values.

## Figures and Tables

**Figure 1 nanomaterials-13-02092-f001:**
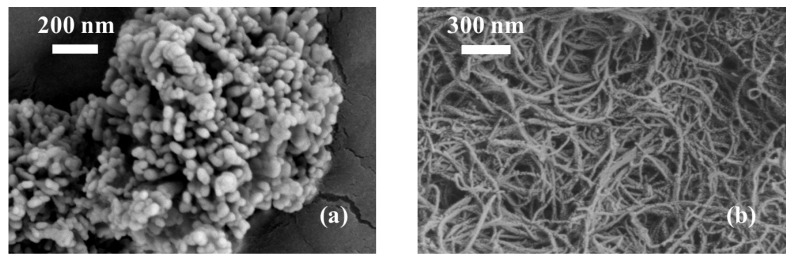
SEM images: (**a**) copper oxide nanoparticles; (**b**) multiwalled carbon nanotubes.

**Figure 2 nanomaterials-13-02092-f002:**
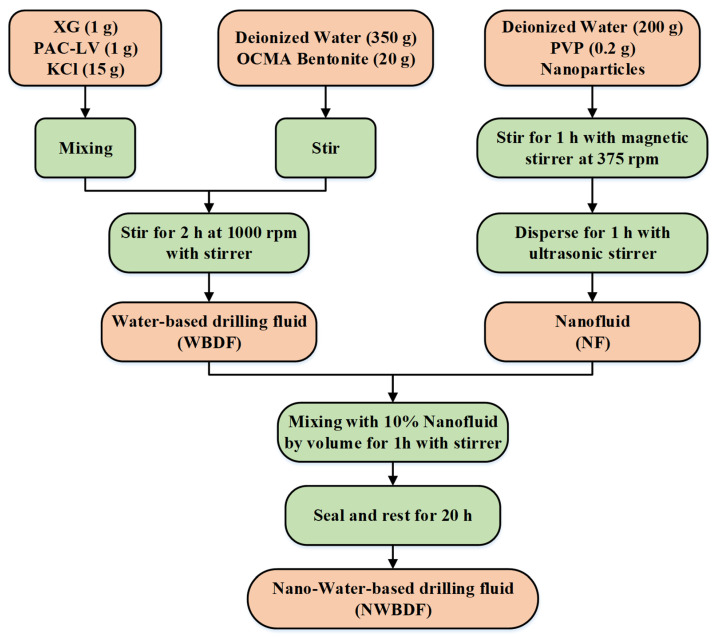
The mixing design of the NWBDFs.

**Figure 3 nanomaterials-13-02092-f003:**
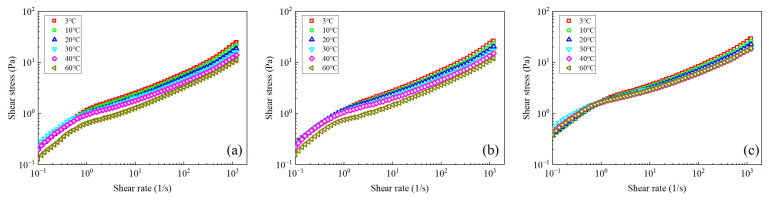
Variation in the shear stress with the shear rate at different temperatures: (**a**) WBDF without NPs; (**b**) NWBDF with 0.15 wt% nano-CuO; (**c**) NWBDF with 0.15 wt% MWCNTs.

**Figure 4 nanomaterials-13-02092-f004:**
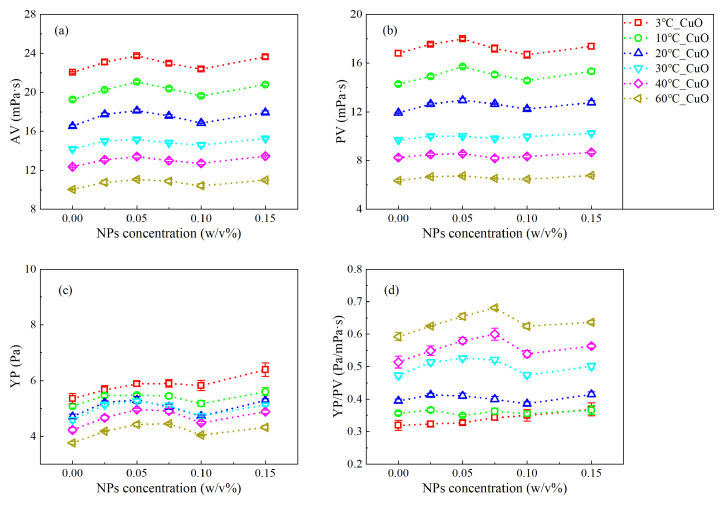
Change in the rheological properties of the NWBDFs at different temperatures with the addition of various concentrations of CuO NPs: (**a**) AV varies with different nanoparticle concentrations and different temperatures; (**b**) PV varies with different nanoparticle concentrations and different temperatures; (**c**) YP varies with different nanoparticle concentrations and different temperatures; (**d**) YP/PV varies with different nanoparticle concentrations and different temperatures.

**Figure 5 nanomaterials-13-02092-f005:**
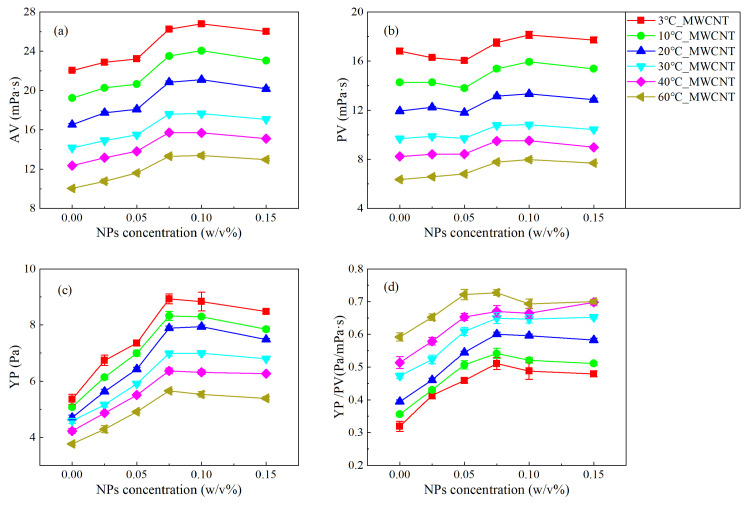
Change in the rheological properties of the NWBDFs at different temperatures with the addition of various concentrations of MWCNTs: (**a**) AV varies with different nanoparticle concentrations and different temperatures; (**b**) PV varies with different nanoparticle concentrations and different temperatures; (**c**) YP varies with different nanoparticle concentrations and different temperatures; (**d**) YP/PV varies with different nanoparticle concentrations and different temperatures.

**Figure 6 nanomaterials-13-02092-f006:**
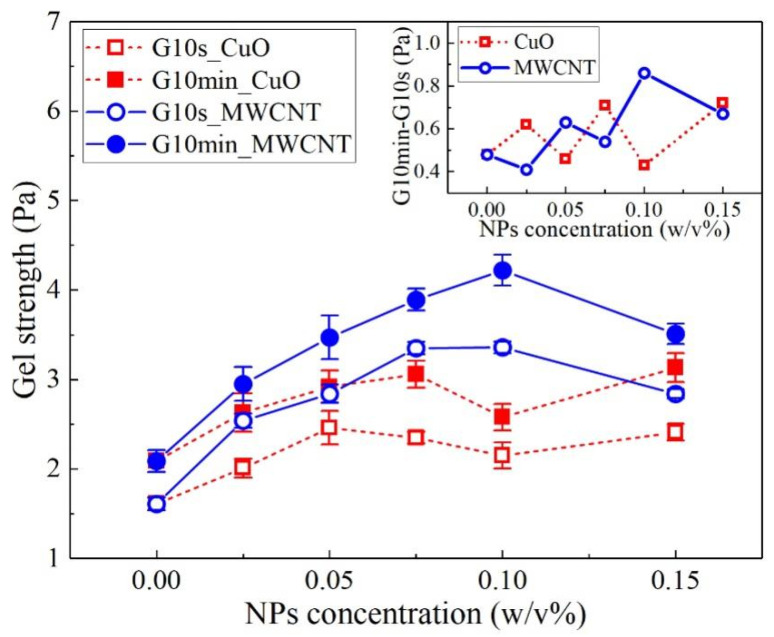
Changes in the gel strength of the two NWBDFs with the concentration of the NPs.

**Figure 7 nanomaterials-13-02092-f007:**
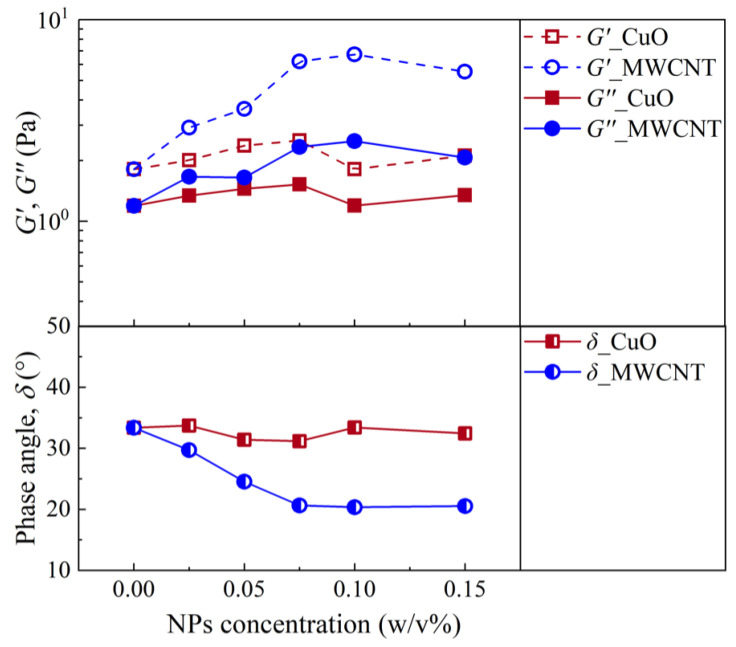
Changes in the modulus and phase angle with the concentration of the NPs at a frequency of 1 Hz for the two NWBDFs at 20 °C.

**Figure 8 nanomaterials-13-02092-f008:**
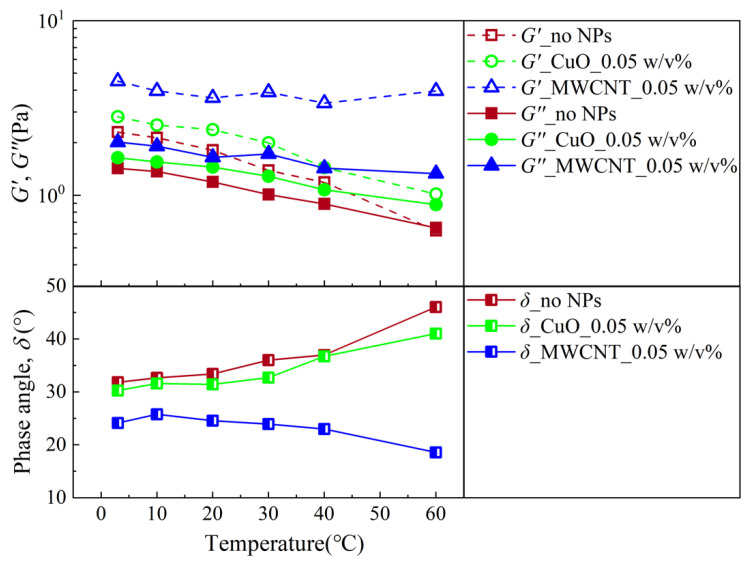
Changes in the modulus and phase angle with the temperature at a frequency of 1 Hz for the two NWBDFs at a NP concentration of 0.05 w/v%.

**Figure 9 nanomaterials-13-02092-f009:**
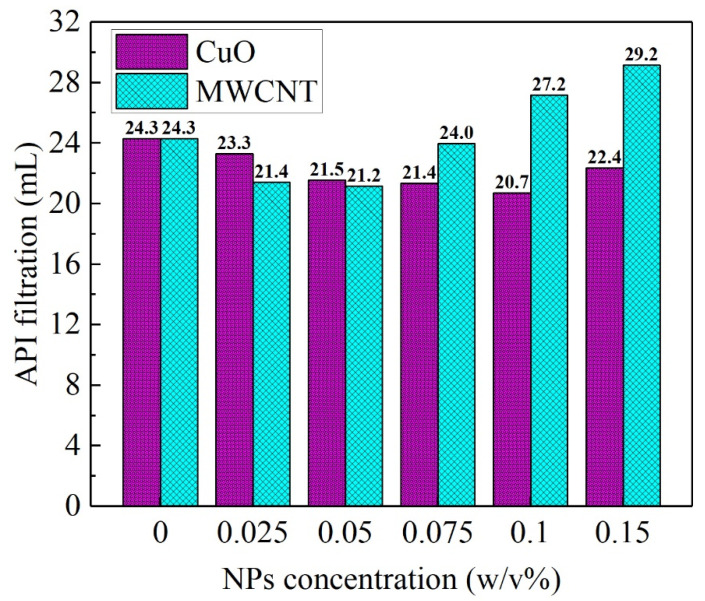
API filtration of the two NWBDFs’ change with the concentration of the NPs.

**Figure 10 nanomaterials-13-02092-f010:**
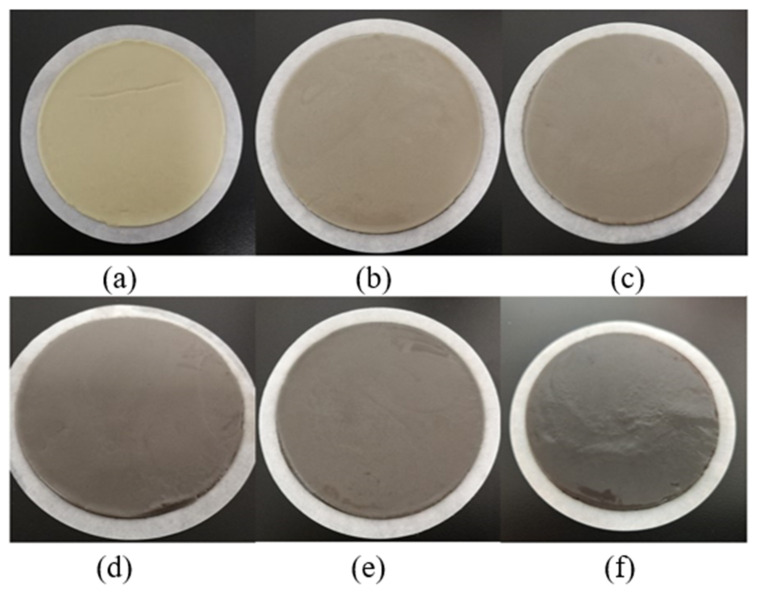
Filter cake obtained from the filtration experiments with the CuO-NWBDF and different NP concentrations: (**a**) no NPs, (**b**) 0.025 w/v%; (**c**) 0.05 w/v%, (**d**) 0.075 w/v%; (**e**) 0.1 w/v%; (**f**) 0.15 w/v%.

**Figure 11 nanomaterials-13-02092-f011:**
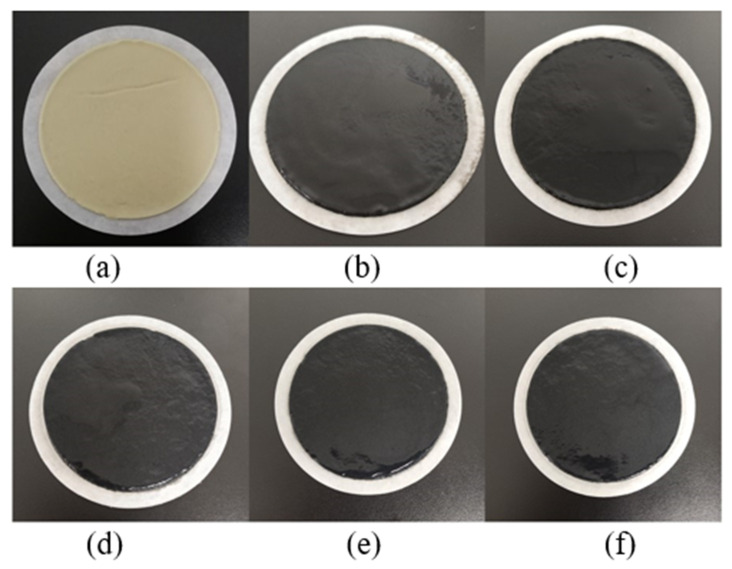
Filter cake obtained from the filtration experiments with the MWCNT-NWBDF and different NP concentrations: (**a**) no NPs; (**b**) 0.025 w/v%; (**c**) 0.05 w/v%; (**d**) 0.075 w/v%; (**e**) 0.1 w/v%; (**f**) 0.15 w/v%.

**Figure 12 nanomaterials-13-02092-f012:**
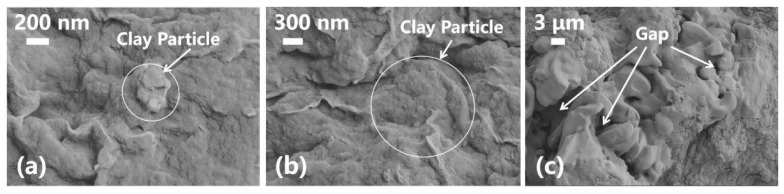
SEM photographs of filter cake of WBDF without NPs: (**a**) 200 nm, (**b**) 300 nm; (**c**) 3 μm.

**Figure 13 nanomaterials-13-02092-f013:**
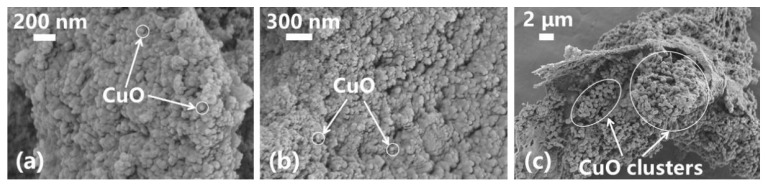
SEM photographs of filter cake of CuO-NWBDF: (**a**) 200 nm, (**b**) 300 nm; (**c**) 2 μm.

**Figure 14 nanomaterials-13-02092-f014:**
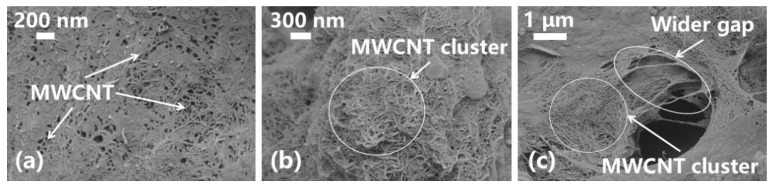
SEM photographs of filter cake of MWCNT-NWBDF: (**a**) 200 nm, (**b**) 300 nm and (**c**) 1 μm.

**Figure 15 nanomaterials-13-02092-f015:**
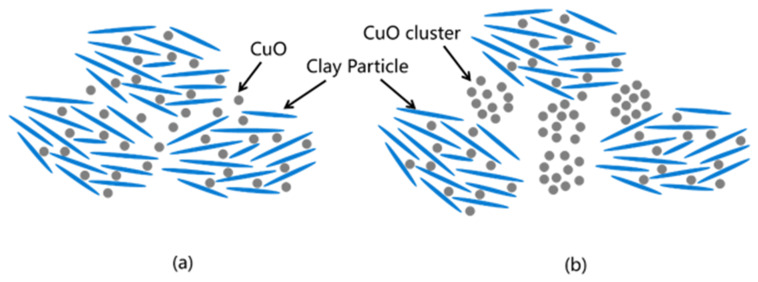
Schematic diagram of the influence mechanism of CuO nanoparticles on filtration: (**a**) Moderate concentration of CuO; (**b**) Excessive concentrations of CuO.

**Figure 16 nanomaterials-13-02092-f016:**
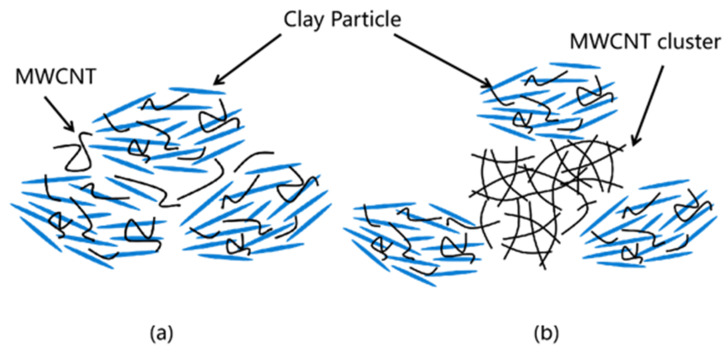
Schematic diagram of the influence mechanism of MWCNTs on filtration: (**a**) Moderate concentration of MWCNT; (**b**) Excessive concentrations of MWCNT.

**Figure 17 nanomaterials-13-02092-f017:**
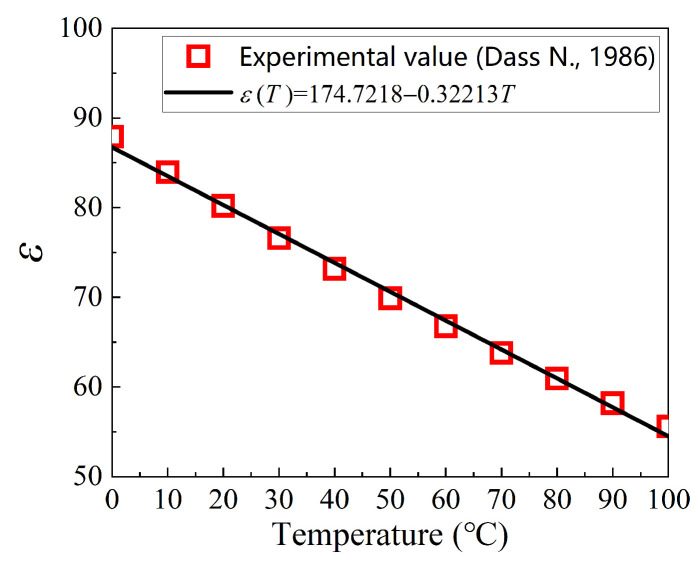
Change in the relative dielectric constant of water with temperature (experimental data from Dass [43].

**Figure 18 nanomaterials-13-02092-f018:**
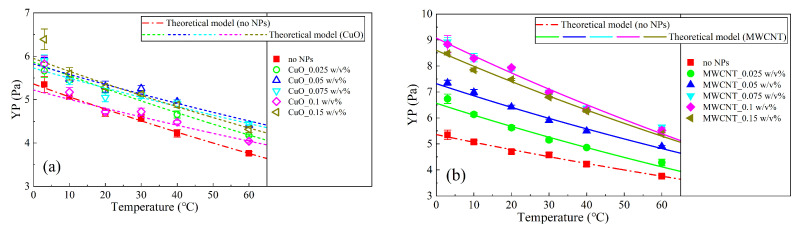
YP changes of temperatures with different nanoparticles (**a**) CuO and (**b**) MWCNT and different NP concentrations. The lines indicate the theoretical curves obtained by fitting Equation (12).

**Figure 19 nanomaterials-13-02092-f019:**
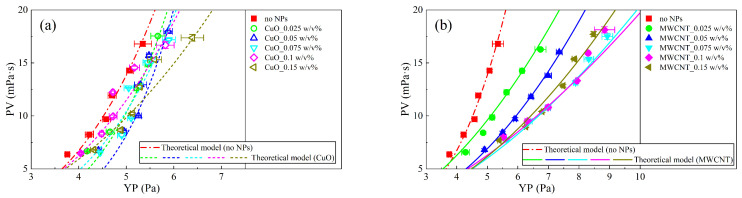
PV changes of YP with different nanoparticles (**a**) CuO and (**b**) MWCNT and different NP concentrations. The lines indicate the theoretical curves obtained by fitting Equation (14).

**Figure 20 nanomaterials-13-02092-f020:**
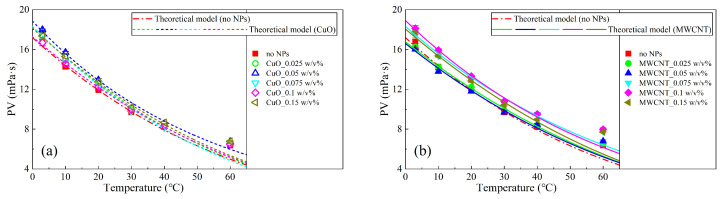
Experimental and theoretical curves of PV versus temperature for different nanoparticles (**a**) CuO and (**b**) MWCNT.

**Table 1 nanomaterials-13-02092-t001:** Application status of NPs in the drilling fluid field.

Author	Year	Drilling Fluid Type	Type and Size of NPs	Modification Effects	Reference
Kosynkin et al.	2012	WBDF (Bentonite)	Graphene Oxide	Improved filtration property	[18]
Abdo et al.	2013	WBDF (Montmorillonite)	Palygorskite (10–20 nm)	Improved rheological properties	[19]
William et al.	2014	WBDF (Bentonite)	CuO and ZnO (<50 nm)	Improved temperature and pressure resistance, increased viscosity	[20]
Barry et al.	2015	WBDF (Bentonite)	${\mathrm{Fe}}_{3}O_{4}\;(3–30\;\mathrm{nm}$)	Improved rheological and filtration properties	[21]
Ismail et al.	2016	WBDF	MWCNT (21 nm) ${\mathrm{SiO}}_{2}\;(12\;\mathrm{nm}$)	Improved rheological and filtration properties	[22]
Dargahi-Zaboli et al.	2017	OBDF (Oil–Water Ratio 70:30)	${\mathrm{SiO}}_{2}\;(100\;\mathrm{nm}$)	Improved thermal stability	[23]
Perween et al.	2018	WBDF (Bentonite)	${\mathrm{ZnTiO}}_{3}\;(200–500\;\mathrm{nm}$) (20–100 nm)	Improved heat resistance and filtration properties	[24]
Wang et al.	2018	WBDF (Bentonite)	${\mathrm{Fe}}_{3}O_{4}\;(10–20\;\mathrm{nm}$)	Improved rheological and filtration properties	[25]
Ghasemi et al.	2018	OBDF (Oil–Water Ratio 10:90)	${\mathrm{Al}}_{2}O_{3}\;(20\;\mathrm{nm}$) ${\mathrm{TiO}}_{2}\;(60\;\mathrm{nm}$)	Improved rheological and filtration properties	[26]
Elochukwu et al.	2018	WBDF (Bentonite)	Nanopolystyrene (25 nm)	Improved rheological and filtration properties	[27]
Dejtaradon et al.	2019	WBDF (Bentonite)	CuO and ZnO (50 nm)	Improved rheological and filtration properties	[28]
Saboori et al.	2019	WBDF (Bentonite)	CuO/PAM	Improved rheological and filtration properties	[29]
Sajjadian et al.	2020	WBDF (Bentonite)	${\mathrm{TiO}}_{2}\;(40\;\mathrm{nm}$) ${\mathrm{SiO}}_{2}\;(40\;\mathrm{nm}$) MWCNT (40 nm)	Improved rheological and filtration properties	[30]
Novara et al.	2021	WBDF (Bentonite)	SiO_2_ and Al_2_O_3_	Improved rheological and filtration properties	[31]
Mirzaasadi et al.	2021	WBDF	SiO_2_	Improved rheological properties	[17]
Mikhienkova et al.	2022	OBDF (Oil–Water Ratio 70:30)	${\mathrm{SiO}}_{2}\;(80\;\mathrm{nm}$)	Improved rheological and filtration properties	[32]
Cheraghi et al.	2022	WBDF	${\mathrm{Al}}_{2}O_{3}\;(30\;\mathrm{nm}$) ${\mathrm{SiO}}_{2}\;(30\;\mathrm{nm}$) ${\mathrm{TiO}}_{2}\;(20\;\mathrm{nm}$)	Improved thermal stability	[33]

**Table 2 nanomaterials-13-02092-t002:** Materials used to prepare the WBDFs.

Deionized Water	OCMA-Grade Bentonite	XG	PAC-LV	KCl
350 g	20 g	1 g	1 g	15 g

**Table 3 nanomaterials-13-02092-t003:** Nanofluids’ formulation.

Number	Deionized Water (g)	PVP (g)	CuO (g)	MWCNTs (g)	Concentration of NPs (*w*/*v*%)
1	200	0.2	−	−	0
2	200	0.2	0.5	−	0.25
3	200	0.2	1.0	−	0.50
4	200	0.2	1.5	−	0.75
5	200	0.2	2.0	−	1.00
6	200	0.2	3.0	−	1.50
7	200	0.2	−	0.5	0.25
8	200	0.2	−	1.0	0.50
9	200	0.2	−	1.5	0.75
10	200	0.2	−	2.0	1.00
11	200	0.2	−	3.0	1.50

**Table 4 nanomaterials-13-02092-t004:** Parameters adopted during fitting of YP using Equation (12) (in the case of δ′=0.01).

Type and Concentration of NPs	$\tau_{vdw}$ (Pa)	$K_{e}\;\left(Pa\;\cdot \;nm\right)$	$\delta^{\prime }\;\left(nm\right)$	R^2^
no NPs	1.1307 × 10^7^	110.091	0.01	0.99115
CuO—0.025 w/v%	1.2290 × 10^7^	114.649	0.01	0.98433
CuO—0.05 w/v%	1.6089 × 10^7^	117.405	0.01	0.85988
CuO—0.075 w/v%	1.6162 × 10^7^	118.500	0.01	0.8083
CuO—0.1 w/v%	1.6242 × 10^7^	110.078	0.01	0.77229
CuO—0.15 w/v%	1.4828 × 10^7^	116.694	0.01	0.93545
MWCNT—0.025 w/v%	3.0072 × 10^7^	172.386	0.01	0.9798
MWCNT—0.05 w/v%	2.4284 × 10^7^	170.259	0.01	0.97329
MWCNT—0.075 w/v%	3.8621 × 10^7^	229.217	0.01	0.97259
MWCNT—0.1 w/v%	4.9517 × 10^7^	256.460	0.01	0.95113
MWCNT—0.15 w/v%	4.1711 × 10^7^	231.160	0.01	0.98832

**Table 5 nanomaterials-13-02092-t005:** Parameters adopted for fitting of the changes in PV with YP using Equation (14).

Type and Concentration of NPs	P	m	R^2^
no NPs	0.07654	3.22918	0.8665
CuO—0.025 w/v%	0.01308	4.14329	0.91483
CuO—0.05 w/v%	0.00293	4.92964	0.88079
CuO—0.075 w/v%	0.04204	3.42462	0.84135
CuO—0.1 w/v%	0.14190	2.72975	0.98357
CuO—0.15 w/v%	0.24815	2.28716	0.88557
MWCNT—0.025 w/v%	0.45040	1.90060	0.96969
MWCNT—0.05 w/v%	0.21757	2.14809	0.9961
MWCNT—0.075 w/v%	0.35275	1.76141	0.96033
MWCNT—0.1 w/v%	0.44156	1.65033	0.91679
MWCNT—0.15 w/v%	0.27534	1.93192	0.94567

## Data Availability

The data presented in this study are available on request from the first author.

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
