# Peer review of "Effect of Nanoparticles on Rheological Properties of Water-Based Drilling Fluid"

_nanomaterials, 2023, doi:10.3390/nano13142092_

Round 1

Reviewer 1 Report

The authors study the effect of using two different nanoparticles on the rheological and filtration properties of a bentonite based drilling fluid.

Similar works have been published in the past as nanoparticles do affect such properties. Additional works to be published should present major effects of nanoparticles on such properties, which however this study does not have.

It can be considered a good engineering study with respect to the experimental results presented. The modelling effort that has been done and presented in the manuscript is a good addition, however it should be tested on samples with greater variety of rheological properties than of the present study. In this study the yield point varies between 3.5 and 5.5 Pa only while the PV varies between 5 to 16 cP and such a drilling fluid could be used in drilling only for a very very limited cases. In other words, a greater variation of rheological properties would be expected to be studied. Also wehter a model is a good fit one does not determine it visually but uses statistical indicators, e.g. correlation coefficient and sum of errors squared.

Furthermore, the authors opted to study AV, YP and PV – and not looking at the full rheograms of the samples where one could have a better idea about the rheological behavior of such samples, i.e. the ‘true’ yield stress of the samples and effect of nanoparticles and whether the samples are shear thinning and if nanoparticles affect such behavior.

With respect to the effect on filtration, the reader sees a very minimal effect with a maximum reduction of less than 15% at the ‘optimal’ concentration of nanoparticles and with the standard API filterpress, whilst many studies which tried to see effect of nanos on filtration properties did that either with HPHT filter press or after thermal aging of the samples, i.e. after exposing the samples to high temperatures.

SOME ADDITIONAL COMMNETS TO THE AUTHORS

When presenting effects of nanos, not a good practice to put all in one figure – very crowded figure – the authors should separate the results of one nanoparticle and present the results of the second nanoparticle in another figure.

The <15% effect on filtration cannot be considered ‘very effective’.

In Figs. 13-14-15, with the SEM pictures, we do not know the % of nanos. Good to indicate % naos plus to the have cross-checking of SEM pictures with the ‘optimal’ and worst performance of nanos to see if any difference. Of course one should stress the fact that SEM shows a very small footprint of the filtercake but some info can be gathered.

good

Reviewer 2 Report

The paper "Effects of Nanoparticles on rheological properties of water-based drilling fluid" represents a thorough investigation of the impact on the rheological behavior of WBDFs of two kinds of nanofillers, Nano-copper oxide (CuO) and multi-walled carbon nanotubes (MWCNT).

It was found that nanoparticles may improve the rheological properties and reduce the filtration of mud cake. These effects are more evident when adopting MWCNTs, which demonstrates that they are more suitable for various applications. In addition, temperature dependence on the rheology is more apparent at higher filler concentrations.

A theoretical model based on the DLVO theory was developed to predict the yield point (YP) and plastic viscosity (PV) of NWBDF as a function of temperature.

The article's objective is clear, the presentation well-structured, and the results well-arranged. The bibliographic references in the introduction help the reader correctly put the contribution within the current state of the art.

Overall, this article deserves publication.

There are a few aspects that require attention.

1.       Xanthan Gum was used as a viscosifier and stabilizer in the system. The addition of xanthan gum significantly affects rheological properties, often representing the dominant factor in many cases. This aspect has to be, at least, discussed in more detail.

2.       A shearing time equal to 10 s-1 at low shear rates (e.g., at 0.1 s-1) may be insufficient to reach equilibrium. This is especially true when dealing with anisotropic and/or structured materials, as in the case at hand. Please justify this small value for acquisition.

3.       It could be useful to report examples of complete viscosity curves, even in the supporting material. Indeed, one cannot appreciate the existence of a shear thinning behaviour from the graphs reported in the manuscript.

Some minor aspects:

1.       More details on the rheometer could be useful (it is not reported if it is a stress- or strain-controlled rheometer)

2.       Row 336: The authors assert that the parameters tau_vdw, K_e, and delta were estimated by fitting. There was no description of this fit in the text.

3.       The Author Contributions section is empty. Please fill it.

4.       References 36 and 38 are the same. Remove one of them.

Reviewer 3 Report

The authors have desgined new water-based drilling fluids and have presented a quite complete rheological characterization according to temperature and the quantity of nanoparticles added. I think that the manuscript deserves publication in Nanomaterials after some minor suggestions:

Minor

·         English should be revised. For example, most of the time that the authors refer to fluid, I think that it should be fluids. The authors also use won’t instead of will not. This is written English, and we all make mistakes, but a revision of language is required.

·         Figure 2. I think that this flow chart is more likely to be a mix design.

·         “The shearing time in each shear step is 10 ? in order for the system to achieve an equilibrium shearing state before the rheological data is recorded”. I disagree with this sentence. The steady state might not be reached at this step time at low frequencies.

·         The information given in figures 3, 4, 5 and 6 is a bit trivial. In my opinion the authors spend too much time in explaining expected results.  I suggest merging the four figures and give a global explanation of the rheological behavior.

·         With SAOS data, have the authors represented the complex viscosity versus temperature? If they do so, an Arrhenius-like tendency might appear (10.1016/j.molliq.2022.120446).

·         The data used in Figure 18 are not clearly explained. How similar are the systems in both articles? Is this general for any aqueous mixture?

·         I cannot see very clearly the data in the last figure. I suggest the authors to include a table with the quality of the fittings, such as R2.

 Typos

·         Line 23: wildly or widely?

·         Table 1. I think that this table should be rearranged, but I guess that this will be done in the print-prove. Anyway, keep the idea in mind.

Although the manuscript is completely understandable, the style is not the required for a scientifc publication.

Round 2

Reviewer 1 Report

in the abstract, please correct, line 14, the apparatus was a Low Pressure Low Temperature filtration apparatus, not medium

in abstract, line 16, it is mentioned '...which not only improves the rheology property..' - you need to rephrase and clarify what do you mean by 'improves'

The effects of addition of nanos, both on rheological behavior and on filtration are small, on would expect much higher impact in order to have research results of high quality. Still, the article is considered an engineering study.

Need minor improvements
